# Redistributing foundation model logits for better domain generalization in low-shot classification

## Abstract

Confidence calibration is an emerging challenge in real-world decision systems that repurpose foundations models for downstream vision classification tasks. Due to various reasons such as inherent biases in contrastive pre-training, frozen embeddings and neural collapse in CLIP when post-training, logit scores on the CLIP head remain large irrespective of whether the image-language pairs reconcile. Ideally, they should be proportional to the amount of that reconciliation. This paper adaptively regulates that reconciliation. We propose a penalty incorporated into the oss objective that penalizes incorrect classifications whenever one is made during adaptation by moving an amount of log-likelihood to the true class commensurate to the relative amplitudes of the two likelihoods. We refer to the latter as the *confidence misalignment penalty (CMP)*. Extensive experiments on 12 vision datasets and 5 domain generalization datasets supports the calibration performance of our method against stat-of-the-art. CMP outperforms the benchmarked prompt learning methods, demonstrating average improvement in Expected Calibration Error (ECE) by average 6.01%, 4.01 % at minimum and 9.72% at maximum.

## 1 Introduction

Foundation models such as CLIP Radford et al. (2021), ALIGN Jia et al. (2021), and Flamingo Alayrac et al. (2022), have demonstrated remarkable performance across a wide range of tasks by leveraging pre-trained multimodal representations, including few-shot and zero-shot classification with tunable language prompts. However, when these foundation models are used for downstream tasks (e.g., classification or retrieval tasks), the setup is observed to frequently lead to poor calibration and produces overconfident misclassified predictions Ye et al. (2023); Tu et al. (2024a) on the downstream task. This leads to problems such as an eventual reduction in generalization and the ascribing of undesirably high confidence to misclassified and out-of-distribution examples. This has in practice resulted in decisions with ethical, equity and health & safety consequences Firoozi et al. (2023), thereby eroding the trustworthiness of this iteration of AI for the general public. This could be viewed as a special case of *misalignment*, where the behavior of foundation models deviates from its objective, or expectation of users or society. The major cause for this is post-training on data without an alignment of the downstream task to the often-undisclosed distribution of data in pre-training or to specific guideline or safety conditions Ouyang et al. (2022); Achiam et al. (2023). We understand the problem to have emerged from a mix of factors: 1) a discounting of the maximum entropy theorem (Sec. 3 Theorem 3.1), 2) neural collapse at post-training time (Sec. 3), and 3) upstream non-plasticity embedding under low-shot exposure to downstream data. We further specify the problem of post-training under the reality of distributional divergence. Model uncertainty often arises under distribution or domain shift, such as due to bad weather condition or poor visibility, then failing gracefully is a more autonomous option than to transferring control to a fallback system (i.e human) Filos et al. (2020). This work tunes the logits to be *less wrong when misclassifying*. In this contribution, we gather the amount of confidence put on the modal CLIP logit when a misclassification has occurred, reassign the difference to the correct text-image pair, therefore explicitly forcing parameter updates to fix the most broken part of the prediction distribution for a given example.

The work subscribes within the body of work on model uncertainty that is championed by Guo et al. (2017)'s model calibration with a focus on the empirical distribution of the predictions $\hat{y}$. It explores the problem from first principles of modeling the output distribution, and fixes a central problem that increases a more realist estimate of confidence, and increases system trust.

**Contributions**

1. We propose a loss component that suppresses CLIP's confidence on mismatched language–image pairings by moderating the posterior flow whenever an incorrect class attains a higher peak than the true class.

2. We show through extensive experiments that CMP consistently improves both accuracy and calibration across few-shot and distribution-shift benchmarks.

**Illustrative example**

We illustrate undue confidence attribution to competing classes using one class $0$ (Plane) in CIFAR10 as a toy example. The top left quadrant of Fig. 1 shows confidence by definition of logits, where classes besides class $0$ have reasonably high bars, most notably class $8$ (Ship) and others. On its right is the equivalent logit vector for our method. This result carries over upon aggregation over the entire testing set, as shown in the respective quadrants on the bottom row, where correctly classified examples (in blue) versus misclassifications (in yellow). Our penalty introduces distinct variation between the highly-confident correct predictions and the low-confidence wrong predictions.

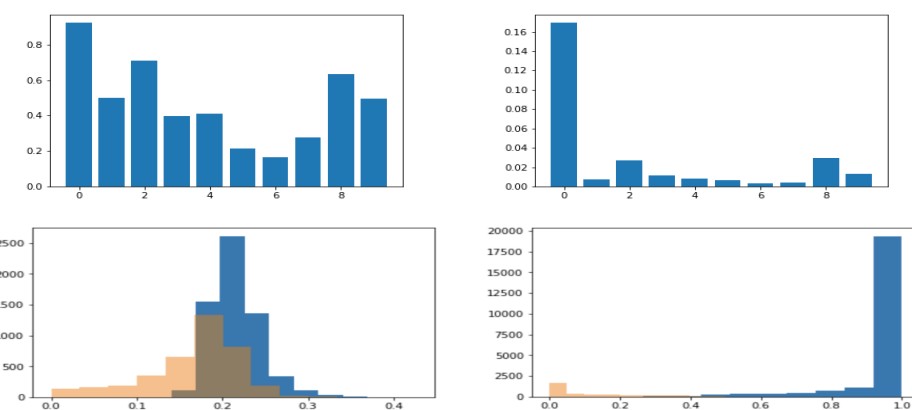

Figure 1: Confidence calibration of CLIP in low-shot (4-shot) settings for CIFAR10 dataset produced using the CoOp pipeline. The top row shows the confidence measurement for class 0 with contrastive loss (left) and confidence misalignment penalty loss (right). The bottom row denotes the confidence level as histogram for correct and incorrect predictions over test set where misalignment penalty shows improvement in calibration by showing logits being less confused.

## 2 RELATED WORK

**Vision-language foundation models.** Foundation models open new horizons for machine learning tasks, particularly in supervised learning Radford et al. (2021); Jia et al. (2021). Foundation models like CLIP can be easily adapted for downstream tasks. To further improve, downstream task classification, several efficient methods are available, including parameter-efficient prompt tuning (PEFT) Zhou et al. (2022b), prompt tuning, vanilla fine-tuning Khattak et al. (2023), adapter tuning Zhang et al. (2021), and CLIP fine-tuning Gao et al. (2024).

The CLIP foundation model Radford et al. (2021), consist of an image encoder $f(x)$ and a text encoder $g(\mathbf{t})$. Both encoders are jointly trained using contrastive loss. The pre-trained CLIP provide support of zero-shot learning which can then be used for downstream tasks like *prompt learning* using fixed or hand-crafted prompts. Context Optimization (CoOp) Zhou et al. (2022b) introduced learnable

task-specific prompts by replacing the manually designed prompts. CoOp provides two different prompt implementation i.e. *Unified context vectors* where prompts shared across classes and *Class-specific context vectors* provides individual prompts per class. But admittedly, CoOp struggles with generalization to unseen classes and distribution shift. Conditional Context Optimization (CoCoOp) Zhou et al. (2022a) generates dynamic prompts using light neural network adaptive to input image. The generated prompt are instance-specific dynamic prompts capable to address overfitting. Prompt distribution learning (ProDA) Lu et al. (2022) models a distribution of diverse prompts instead of individual prompts to generalise well to downstream tasks. ProDA learns a Gaussian distribution of classifiers weights drawn from diverse prompts. Diversity is introduced by varying class-token position in prompt.

**Confidence calibration in deep learning models.** Extensive research focuses on confidence calibration methods to ensure model accuracy is aligned with its predicted confidence scores. These methods includes regularization based approaches, such as implicit regularization Ross & Dollár (2017) Baloch et al. (2019), $L_2$ regularization Guo et al. (2017) , and entropy regularization Pereyra et al. (2017) to align the model predicted accuracy with confidence. Some augmentation-based approaches, such as label-smoothing Müller et al. (2019) and mix-up training methods Thulasidasan et al. (2019); Zhang et al. (2022) are also explored for confidence calibration in deep learning models. Mukhoti et al. (2020) use focal loss to uniformly down-weight the confidence prediction for all samples while $CMP$ which penalize misclassified only. Liu et al. (2022) applies margin smoothing by assuming uniform label noise and requires tuning margin hyperparameters. $CMP$ automatically adapts penalty strength via relative logit amplitudes, requiring no margin tuning.

**Confidence calibration in foundation models.** Pandey et al. (2024) addresses the issue of under-confidence in pre-trained foundation models when these models are fine-tuned for downstream tasks in few-shot settings. Similarly, Murugesan et al. (2025) highlights the miscalibration problem in CLIP-based adaptive approaches. Their proposed method empirically demonstrates that CLIP-based adaptive approaches, such as prompt learning, adapters, parameter-efficient fine-tuning (PEFT), and test-time adaptation, are prone to miscalibration in zero-shot/few-shot settings under distribution shifts. Furthermore, Tu et al. (2024b) investigates the prediction uncertainty of the foundation model CLIP for image classification tasks, focusing on variations in visual factors.

In this work, we propose a regularization-based approach CMP which focus on low-shot regimes where validation set is infeasible to ensure that the predictive probabilities of foundation models align with confidence scores in a prompt-learning setting under distribution shifts.

## 3 METHOD

A fundamental issue with the otherwise highly effective pairing of softmax-based classifiers and cross-entropy loss is prediction overconfidence and underconfidence, where the softmax logits, which are unnormalized log probabilities at the output, tend to overestimate or underestomate the true classification accuracy. Foundation models such as CLIP use contrastive loss to align text and image embeddings Radford et al. (2021). The CLIP loss $\mathcal{L}_c$ can be defined as: $\mathcal{L}_c = \frac{1}{2}\left(\mathcal{L}_{\text{txt}} + \mathcal{L}_{\text{img}}\right)$,

$$\mathcal{L}_{\text{txt}} = -\frac{1}{N}\sum_{i=1}^{N}\log\frac{\exp(\text{sim}(\mathbf{t}_i,\mathbf{i}_i)/\tau)}{\sum_{j=1}^{N}\exp(\text{sim}(\mathbf{t}_i,\mathbf{i}_j)/\tau)}, \mathcal{L}_{\text{img}} = -\frac{1}{N}\sum_{i=1}^{N}\log\frac{\exp(\text{sim}(\mathbf{i}_i,\mathbf{t}_i)/\tau)}{\sum_{j=1}^{N}\exp(\text{sim}(\mathbf{i}_i,\mathbf{t}_j)/\tau)}$$

where, $\text{sim}(\mathbf{t}_i,\mathbf{i}_j)$ is the cosine similarity between text embedding $\mathbf{t}_i$ and image embedding $\mathbf{i}_j$, $\tau$ is a learned temperature parameter, while N is the batch size.

**Theorem 3.1.** *CLIP's contrastive loss for aligning image-text pairs introduces bias to the posterior probabilities $P(\mathbf{t} \mid \mathbf{i};\theta)$ and $P(\mathbf{i} \mid \mathbf{t};\theta)$, pushing them towards low-entropy distributions, thereby conflicting with the Maximum Entropy Principle (MEP). This is reflected in the* sharpness metric $\mathcal{S}$, *defined as:*

$$\mathcal{S} = \frac{\max_j P(\mathbf{t}_j|\mathbf{i})}{\mathbb{E}_{\mathbf{t}_j\sim\mathcal{U}}[P(\mathbf{t}_j|\mathbf{i})]} = N \cdot P(\mathbf{t}_i|\mathbf{i})$$

*where CLIP optimization drives $\mathcal{S} \to N$ (fully peaked), while MEP favors $\mathcal{S} = 1$ (uniform). Furthermore, the entropy $H\big(P(\mathbf{t}|\mathbf{i};\theta)\big)$ is bounded by:*

$$H(P(\mathbf{t}|\mathbf{i};\theta)) \leq \log N - \frac{N}{N-1}\exp\left(-\frac{2\Delta_{sim}}{\tau}\right),$$

*with $\Delta_{sim} = sim(\mathbf{i}, \mathbf{t}_i) - \max_{j \neq i} sim(\mathbf{i}, \mathbf{t}_j)$ is the similarity gap. As the training progresses and $\Delta_{sim} \to \infty$, the bound forces $H(P) \to 0$, directly opposing MEP's principle.*

*Proof.* By definition of the CLIP loss:

$$\mathcal{L}_{\mathrm{C}} = -\frac{1}{2N} \sum_{i=1}^{N} \left[ \log P(\mathbf{t}_i|\mathbf{i}_i) + \log P(\mathbf{i}_i|\mathbf{t}_i) \right],$$

where: $P(\mathbf{t}|\mathbf{i}; \theta) = \frac{\exp(\mathrm{sim}(\mathbf{i},\mathbf{t})/\tau)}{\sum_{j=1}^{N} \exp(\mathrm{sim}(\mathbf{i},\mathbf{t}_j)/\tau)}$. Here, $\mathrm{sim}(\mathbf{i}, \mathbf{t})$ is the cosine similarity, $\tau$ is a learned temperature parameter, and $N$ is the batch size.

The entropy of $P(\mathbf{t}|\mathbf{i}; \theta)$ is:

$$H(P(\mathbf{t}|\mathbf{i}; \theta)) = -\sum_{j=1}^{N} P(\mathbf{t}_j|\mathbf{i}) \log P(\mathbf{t}_j|\mathbf{i}).$$

CLIP's loss minimization forces $P(\mathbf{t}_i|\mathbf{i}) \to 1$ (correct pair) and $P(\mathbf{t}_j|\mathbf{i}) \to 0$ (negative pairs, $j \neq i$), reducing entropy $H(P) \to 0$. This behavior is quantified by entropy bound which is derived from dominance of correct pair similarity:

$$P(\mathbf{t}_j|\mathbf{i}) \leq \frac{\exp(-\Delta_{\mathrm{sim}}/\tau)}{1 + (N-1)\exp(-\Delta_{\mathrm{sim}}/\tau)}, \quad j \neq i$$

leading to the upper bound on entropy.

According to the Maximum Entropy Principle (MEP) Jaynes (1957), the least biased distribution that best represents uncertain data is the one with maximum entropy:

$$P_{\mathrm{MEP}} = \arg\max_P \left\{ -\sum_y P(y|x) \log P(y|x) \right\},$$

which, under normalization ($\sum_j P(\mathbf{t}_j|\mathbf{i}) = 1$), favors a uniform distribution where all outcomes have non-negligible probability.

To quantify the deviation from MEP, we adapt sharpness metric $\mathcal{S}$ inspired by distributional peakedness measures Theodoridis (2015); Guo et al. (2017); Tian (2022):

$$\mathcal{S} = \frac{\max_j P(\mathbf{t}_j|\mathbf{i})}{\mathbb{E}_{\mathbf{t}_j \sim \mathcal{U}}[P(\mathbf{t}_j|\mathbf{i})]} = N \cdot P(\mathbf{t}_i|\mathbf{i})$$

since $\mathbb{E}[P(\mathbf{t}_j|\mathbf{i})] = \frac{1}{N}$ (due to uniform expectation under no assumptions).

Under CLIP optimization:

$$P(\mathbf{t}_i|\mathbf{i}) \to 1 \implies \mathcal{S} \to N \quad \text{(fully peaked)},$$

whereas MEP requires $\mathcal{S} = 1$ (uniform distribution).

Thus, CLIP's loss minimization directly opposes MEP through both metrics:

$$\mathcal{S} \to N \quad \text{and} \quad H(P) \to 0.$$

$\square$

We emphasize that MEP is applied selectively. CLIP's contrastive loss drives all predictions toward peaked distributions ($\mathcal{S} \to N$), but this is only problematic for misclassified examples where high confidence is unjustified. For correct predictions, peaked distributions ($S \to N$) are desirable. CMP therefore applies MEP-inspired regularization only when $P(x, y') > P(x, y)P(x, y')$ (misclassification), preserving appropriate confidence for correct predictions while reducing overconfidence for errors.

However, these models are prone to confidence not being proportionate to predictions Wen et al.. The important remark is that for a wrong prediction $\hat{y}$, any class that is not the true image class $y'$"steals"

probability mass from $y'$. Indeed, even for a correct prediction, this mass is spread to possibly all competing class choices. This amount of lost likelihood gives us a measure of confidence lost in the prediction. We call the quantity, *confidence misalignment* (CM).

For well-calibrated prediction, the sofmax probability $P(x, y)$ assigned to the correct predicted class $y$ must be higher than the incorrect classes $(P(x, y'), y \neq y')$. Therefore, we re-introduce a fraction of likelihood to penalize the contrastive loss, the confidence misalignment penalty (CMP) [1].

$$\text{CMP} = \frac{P(x, y)}{\sum_{y' \neq y \, : \, P(x, y') > P(x, y)} P(x, y')}, \tag{1}$$

where $P(x, y) = \frac{\exp(s(x, y))}{\sum_j \exp(s(x, y))}$ is softmax probability for correct pair and $P(x, y') = \frac{\exp(s(x, y'))}{\sum_k \exp(s(x, y_k))}$ represents the softmax probabilities for all incorrect pairs $(x, y')$ where $y \neq y'$.

**Proposition 3.1.** *CMP $\leq 1$ thereby bounding the confidence by redistributing the probability mass from overconfident incorrect predictions to the correct prediction. (see proof detail B.1)*

The CMP for a batch N can be defined as:

$$\mathcal{L}_{\text{CMP}} = \frac{1}{N} \sum_{i=1}^{N} \frac{P(x_i, y_i)}{\sum_{y' \neq y_i, \, P(x_i, y') > P(x_i, y_i)} P(x_i, y') + \epsilon}. \tag{2}$$

After plugging in this into CLIP loss, the final CLIP loss $\mathcal{L}_{\text{final}}$ is defined as:

$$\mathcal{L}_{\text{final}} = \mathcal{L}_{\text{c}} + \lambda . \mathcal{L}_{CMP} \tag{3}$$

$\lambda$ is calibrating strength of CMP penalty.

**Proposition 3.2.** *The CMP penalty for correctly classified pairs is $\mathcal{L}_{CMP}(x, y) \approx 0$, while for misclassified pairs, $\mathcal{L}_{CMP}(x, y') > 0$. (see proof detail B.2)*

This results in a significant CMP penalty, as it redistributes the confidence from overconfident incorrect predictions. This redistribution cofidence penalty is integrated into CLIP loss which is denoted as $\mathcal{L}_{\text{final}}$. The introduced penalty works dynamically by penalizing misclassified pairs only and leaving the correct one. Next, we make sure that CMP loss is bounded and gradient decays smoothly to ensure stability.

**Proposition 3.3.** *The gradient of the CMP loss $\mathcal{L}_{CMP}$ with respect to logits is bounded and ensures stable optimization. (see proof detail B.3)*

The introduced CMP addresses the critical challenge of miscalibration in foundation models, by redistributing the probability mass from incorrect prediction to correct one. CMP integration into CLIP ensures reliable predictions which are aligned with true probability. Furthermore, theoretical guarantees like confidence bound, gradient stability, convergence and selective penalty makes the proposed method robust for use in real-world applications where trust is in predictions is critical.

## 4 EXPERIMENTS AND RESULTS

### 4.1 EXPERIMENTAL SETUP

**Datasets.** We evaluated our method CMP on 12 standard vision benchmark datasets that includes, Flowers102 Nilsback & Zisserman (2008), StanfordCars Krause et al. (2013), Imagenet Deng et al. (2009), DTD Cimpoi et al. (2014), SUN397 Xiao et al. (2010), CIFAR10 Coates et al. (2011), EuroSAT Helber et al. (2019), OxfordPets Parkhi et al. (2012), UCF101 Soomro (2012), Food101 Bossard et al. (2014), FGVCAircraft Maji et al. (2013), Caltech101 Fei-Fei et al. (2004). For domain generalisation we used ImageNet-1K with its four variants like ImageNet A (Art) Hendrycks et al.

---

[1] We fondly like to refer to the model's reflexive epistemics and do a wordplay with the name of the penalty. Referring to to the social psychologists Dunning and Kruger's work Kruger & Dunning (1999), we *informally* call our loss the Dunning-Kruger Loss (the DKL), reading it as the reflected anagram (i.e. a *guided redistribution*) of KLD, the Kullback-Leibler Divergence, which fixates on a high-bias region around the mode.

(2021b), ImageNet-V2 (V2) Recht et al. (2019), ImageNet R (Real) Hendrycks et al. (2021a) and ImageNet S (Sketch) Wang et al. (2019) to ensure its performance.

**Implementation details.** Our experiment is in two parts. In first part we evaluate CMP performance for prompt learning methods. To achieve this, We used pre-trained CLIP (ViT-B16) Radford et al. (2021) model in prompt learning experiments. In second part, we use CLIP (ViT-B16) and (ResNet-50) as backbone while comparing CMP performance with CLIPCalib Murugesan et al. (2025) [2] in domain generalization. To check the performance of CMP we performed following sets of experiments.

- **Evaluating calibration in CLIP-based models.** We conduct an experiment to analyze the overconfidence in foundation model when adapted for downstream vision classification tasks. The results are shown in Figure 1 and a detail discussion can be found in sec 5.

- **Improving prompt learning calibration with CMP integration.** To evaluate the performance of our penalty integration into prompt learning methods, we performed prompt learning with and without penalty integration. The results are reported in tabular form and detail discussion is provided in sec 5.

- **CMP integration enhances calibration in domain generalization.** To analyze the robustness of CMP, we performed experiment in domain generalization settings. The results are presented in tabular formate along with detail discussion in section 5 and appendix C.

- **CMP improvement against benchmarks.** To check the calibration performance of CMP in comparison to benchmarking, we perform detail experiments with two backbone (ResNet-50 and ViT-B16). The results are discussed in section 5 and appendix C.

Due to spatial constraint more experimental detail can be found in appendix C.

**Baselines.** We compare CMP performance with existing prompt learning methods, such as CoOp Zhou et al. (2022b), CoCoOp Zhou et al. (2022a), ProDA Lu et al. (2022), MaPLe Khattak et al. (2023), and KgCoOp Yao et al. (2023).

**Benchmarks.** We compared CMP confidence normalization performance against standard CLIP in zero-shot setting, with zero-shot calibration method CLIPCalib Murugesan et al. (2025) and , C-tpt Yoon et al. (2024), B-PEFT Pandey et al. (2024).

**Evaluation metrics.** To measure the confidence calibration of CMP, we used three evaluation metrics Expected Calibration Error (ECE) Guo et al. (2017), Adaptive Calibration Error (ACE) Guo et al. (2017) and Maximum Calibration Error (MCE) Nixon et al. (2019) and accuracy in our work.

**Low-shot settings:** We evaluate our method CMP in low-shot settings (1-8 shots per class) to assess the generalization and calibration with limited data. The detailed results are given in appendix Table 6 and Table 7. The results show that CMP maintains robust performance compared to baseline prompt learning methods, in low-shot regime.

## 5 DISCUSSIONS

In our work, we evaluated the effectiveness of CMP calibration using ECE, ACE, MCE and generalization of CMP using acc metrics. But due to space constraint we reported ECE results in this section. The remaining metrics results can be found in C.

**Calibration performance of CLIP-based adaptive models.** Figure 1 shows confidence calibration of CLIP in zero-shot settings for CIFAR10 classification task. The top row (*left*) in figure shows contrastive loss result without integration of CMP into loss, it has been observed that the confidence distribution for predicting class 0 is spread across all incorrect classes. This indicates overconfidence of the model as the model logits are less focused. The top row (*right*) shows results of CMP loss for zero-shot tasks, it is shown in figure that the confidence distribution is concentrated near lower values of confidence. This indicates that CMP reduces the overconfidence for incorrect prediction.

---

[2]The CLIPCalib prompt learning comparison results are reproduced by using their code published in paper which is available at `https://github.com/Bala93/CLIPCalib`.

Table 1: CMP average calibration performance across 11 datasets for downstream tasks in 8-shot setting. the row with + shows our method combined with existing prompt learning method. The ↓ represent the smaller value is better, while the ↑ in accuracy (Acc) denotes the larger value is significant. **Bold** values shows more significant result.

| Method | Acc(↑) | ACE(↓) | MCE(↓) | ECE(↓) |
|---|---|---|---|---|
| CLIP | 72.7 | 3.51 | 0.89 | 3.47 |
| CoOp | 83.2 | 3.28 | 1.16 | 3.16 |
| CoOp + (CMP) | **84.1** | **3.01** | **0.98** | **2.98** |
| CoCoOp | 73.8 | 3.65 | 0.92 | 3.97 |
| CoCoOp + (CMP) | 72.2 | 3.71 | 1.03 | 4.07 |
| ProDA | 83.4 | 4.29 | 1.33 | 4.13 |
| ProDA + (CMP) | **84.2** | **3.86** | **1.49** | **3.94** |
| MaPLe | 82.9 | 4.63 | 1.49 | 3.19 |
| MaPLe + (CMP) | **84.6** | **4.01** | **1.21** | **2.88** |
| KgCoOp | 75.2 | 4.61 | 1.09 | 4.24 |
| KgCoOp + (CMP) | **76.3** | **4.11** | **0.97** | **4.07** |

Table 2: Comparison of CMP when combined with prompt learning methods in 8-shot setting. **Bold** values represent significant performance. In the left column Acc ↑ demonstrates larger value is significant. In the right column ECE ↓ shows smaller value is significant.

| Method | Acc (↑) | | | | | | ECE (↓) | | | | | |
|---|---|---|---|---|---|---|---|---|---|---|---|---|
| | source | target | | | | | source | target | | | | |
| | ImageNet | R | S | A | V2 | Avg | ImageNet | R | S | A | V2 | Avg |
| CLIP | 65.8 | 70.5 | 47.1 | 48.8 | 61.5 | 57.2 | 1.94 | 4.12 | 4.98 | 7.92 | 3.01 | 5.01 |
| CoOp | 70.3 | 74.0 | 46.3 | 48.6 | 64.2 | 58.2 | 1.32 | 7.83 | 14.6 | 6.77 | 2.98 | 8.01 |
| CoOp + (CMP) | 72.5 | **74.8** | **47.4** | **49.7** | **66.1** | **59.5** | 1.55 | **6.91** | **13.9** | **6.35** | **2.13** | **7.32** |
| CoCoOp | 71.4 | 74.5 | 50.3 | 51.9 | 65.4 | 60.5 | 1.09 | 6.91 | 12.4 | 5.29 | 2.75 | 6.83 |
| CoCoOp + (CMP) | 72.8 | **77.8** | **50.5** | **53.8** | 65.2 | **61.8** | 1.85 | 7.44 | **11.8** | **4.32** | **1.05** | **6.15** |
| ProDA | 69.6 | 70.7 | 54.2 | 54.8 | 66.4 | 61.5 | 1.39 | 7.55 | 14.9 | 6.91 | 3.05 | 8.10 |
| ProDA + (CMP) | 67.3 | 68.9 | 53.6 | 52.9 | **66.7** | 60.5 | 1.47 | **6.01** | **13.6** | 6.95 | **2.73** | **7.32** |
| MaPLe | 72.5 | 72.1 | 54.9 | 49.7 | 64.3 | 60.3 | 1.53 | 3.66 | 5.78 | 13.5 | 2.01 | 6.48 |
| MaPLe + (CMP) | 70.6 | **73.3** | **55.6** | 52.9 | **65.7** | **61.8** | 1.68 | **2.48** | **4.07** | **12.5** | **1.98** | **5.25** |
| KgCoOp | 73.7 | 72.2 | 49.6 | 50.1 | 63.7 | 58.9 | 2.51 | 4.15 | 7.52 | 14.1 | 3.75 | 7.38 |
| KgCoOp + (CMP) | 72.9 | **74.3** | 48.8 | **52.5** | **65.4** | **60.3** | 2.92 | **4.13** | **7.08** | **13.4** | **3.08** | **6.92** |

Table 3: CMP results in comparison with CLIPCalib Murugesan et al. (2025) with ResNet-50 backbone in 8-shot setting. CMP results for ViT-16 backbone implementation are given in the Appendix C. **Bold** value represents significance improvement in CMP performance. In ACC part (↑) denotes larger value better accuracy. while in ECE part (↓) demonstrate smaller value better calibration.

| | Method | UCF101 | Food101 | Caltech101 | Pets | Flowers102 | ImageNet | Cars | Aircraft | SUN397 | DTD | EuroSAT | Avg. |
|---|---|---|---|---|---|---|---|---|---|---|---|---|---|
| ACC (↑) | CLIP | 58.9 | 73.9 | 85.6 | 83.6 | 61.6 | 58.2 | 55.7 | 15.7 | 58.8 | 40.3 | 23.7 | 56.0 |
| | CLIPCalib | 60.4 | 75.2 | 87.1 | 84.3 | 61.8 | 60.7 | 58.1 | 17.2 | 61.1 | 42.0 | 26.6 | 57.7 |
| | C-tpt | 61.2 | 76.3 | 87.5 | 84.4 | 62.0 | 60.9 | 59.5 | 17.0 | 62.4 | 43.2 | 26.0 | 58.1 |
| | B-PEFT | 60.8 | 75.8 | 87.3 | 84.3 | 61.9 | 60.8 | 58.8 | 17.1 | 61.8 | 42.6 | 26.3 | 57.9 |
| | CLIPCMP | **61.9** | **77.3** | **87.8** | **84.5** | **62.1** | 59.3 | **60.9** | 16.8 | **63.2** | **43.7** | 25.4 | **58.5** |
| ECE (↓) | CLIP | 3.57 | 3.46 | 3.68 | 3.42 | 3.76 | 3.52 | 3.46 | 3.44 | 3.62 | 3.68 | 3.42 | 3.47 |
| | CLIPCalib | 3.48 | 3.38 | 3.59 | **3.34** | 3.67 | **3.44** | **3.38** | 3.36 | 3.54 | **3.49** | 3.44 | 3.39 |
| | C-tpt | 3.45 | 3.34 | 3.55 | 3.38 | 3.48 | 3.47 | 3.41 | 3.28 | 3.50 | 3.55 | 3.31 | 3.35 |
| | B-PEFT | 3.46 | 3.36 | 3.57 | 3.36 | 3.58 | 3.46 | 3.40 | 3.32 | 3.52 | 3.52 | 3.38 | 3.37 |
| | CLIPCMP | **3.41** | **3.30** | **3.52** | 3.57 | **3.29** | 3.36 | 3.40 | **3.19** | **3.46** | 3.61 | **3.17** | 3.31 |

The bottom row (*left*) shows confidence level histogram for correct (blue) and incorrect (orange) predictions using contrastive loss. The overlap indicates confusion in model logits, model can not confidently differentiate between correct and incorrect predictions. The bottom row (*right*) represents confidence for correct (blue) and incorrect (orange) prediction using CMP loss. It has been observed that correct predictions (blue) are sharply concentrated near 1.0 while the incorrect predictions (orange) are near to 0.0. This clear separation shows that CLIP model is better calibrated and less confused in zero-shot setting, even without task specific fine-tuning.

Table 4: CMP performance results in comparison with CLIPCalib with (ViT-B16) as backbone for domain genralisation in 8-shot setting.

| | Method | ImageNet-1K variants | | | | | |
| | | source | target | | | | |
| | | ImageNet | R | S | A | V2 | Avg |
| ACC | CLIP | 65.8 | 70.5 | 47.1 | 48.8 | 61.5 | 57.2 |
| | CLIPCalib | 73.8 | 77.1 | 47.9 | 54.5 | 63.5 | 60.7 |
| | C-tpt | 72.0 | 76.0 | 47.1 | 54.0 | 63.1 | 60.0 |
| | B-PEFT | 71.1 | 75.4 | 46.7 | 53.8 | 62.9 | 59.7 |
| | CLIPCMP | 70.2 | 74.8 | 46.2 | 53.5 | 62.7 | 59.3 |
| ECE | CLIP | 1.94 | 4.12 | 4.98 | 7.92 | 3.01 | 5.01 |
| | CLIPCalib | 1.88 | 3.99 | 4.82 | 7.67 | 2.91 | 4.85 |
| | C-tpt | 1.86 | 3.94 | 4.61 | 7.71 | 2.87 | 4.78 |
| | B-PEFT | 1.85 | 3.91 | 4.50 | 7.73 | 2.85 | 4.75 |
| | CLIPCMP | 1.83 | **3.88** | **4.39** | 7.75 | **2.83** | **4.71** |

**Improving prompt learning calibration with CMP integration.** To evaluate the impact of CMP integration into prompt learning methods, we report ECE performance results across 11 datasets for various baselines, both with and without the penalty term, as shown in Table 1.

The results in Table 1 indicate that integrating CMP consistently reduces ECE, thereby improving model calibration. For example, the baseline CoOp achieves an ECE of 3.16, while its penalty-enhanced version (CoOp + CMP) reduces ECE to 2.98, representing a $5.70\%$ improvement. Similarly, the ProDA baseline shows an ECE of 4.13, which decreases to 3.94 with the penalty-based version (ProDA + CMP), reflecting a $4.60\%$ reduction.

Notably, MaPLe demonstrates the highest improvement, with ECE decreasing from 3.19 to 2.88, corresponding to a $9.70\%$ reduction, highlighting the strong impact of CMP for this specific prompt learning method. In the case of KgCoOp, integrating CMP (KgCoOp + CMP) reduces ECE from 4.24 to 4.07, marking a $4.00\%$ improvement.

These results clearly demonstrate the effectiveness of CMP in mitigating overconfidence in predictions. The consistent reductions in ECE, ranging from a minimum of $4.00\%$ to a maximum of $9.70\%$, underline the robustness of CMP in enhancing calibration across diverse prompt learning approaches.

**CMP integration enhances calibration in domain generalization.** To evaluate the calibration performance of CMP integration, we analyze the ECE across domain generalization datasets. The results are reported in Table 2. We discuss the improvement in ECE for each prompt learning method while focusing on source and target domain. It is observed in Table 2, CoOp achieve an average ECE of 8.01 reflecting significant calibration error. By incorporating our penalty to this method the average ECE is reduced to 7.32 representing $8.56\%$ improvement. The ECE improvement shows the effective mitigation of overconfidence in CoOp method, particularly in domain generalizing setting. The reported ECE for CoCoOp without CMP is 6.83, while penalized CoCoOp ECE is reduced to 6.15, yielding $9.8\%$ improvement in calibration error under challenging target domain.

ProDA baseline achieves $8.10\%$ ECE on average, with penalty integration ECE reduced to 7.32 which is $9.60\%$ improvement. The baseline method MaPLe shows an average of 6.48 ECE, which is reduced to 5.25 by integration of our penalty, marking on average $19.0\%$ performance improvement. KgCoOp shows an average ECE of 7.38 as baseline, while penalty based KgCoOp reports 6.92 corresponding $6.20\%$ reduction in ECE.

From the results given in Table 2, CMP consistently improves in ECE across domain generalization datasets with improvement within range ($6.20\%$ - $19.0\%$). The results confirms the effectiveness of CMP in mitigating overconfidence under challenging scenarios of domain generalization settings.

**CMP improvement against benchmarks.** CMP calibration performance evaluation results in comparison with benchmarking method CLIPCalib, C-tpt, B-PEFT and CLIP (baseline) for vision datasets and domain generalization datasets are given in Table 3 and Table 4 respectively.

In Table 3, it can be observed that CLIPCMP achieves an average ECE of 3.31 significantly improving upon the baseline average ECE i.e $4.49\%$ and $2.20\%$ improvement over benchmark CLIPCalib. The

calibration improvement is notable on various datasets like Flowers102 (3.76 to 3.29), EuroSAT (3.42 to 3.17), and Aircraft (3.44 to 3.19), where CMP excels in resolving overconfidence. The results are clearly in favor of our method, indicating the importance of CMP integration into CLIP framework leads to better-calibrated models. The consistent reduction in ECE across datasets demonstrates the robustness of our method.

Table 4 denotes results of CMP calibration results in comparison with CLIP (baseline) and CLIPCalib, C-tpt, B-PEFT benchmark on domain generalization datasets.

The average calibration error reduces from 5.01 to 4.71 when penalty is plugged-in into CLIP framework, showing an improvement of 5.94% as compared to baseline while 2.79% over the baseline in challenging datasets. We have noticed that our method consistently outperform the benchmark across domain generalization datasets. We have also reported the result for CMP by changing the backbone of CLIP which are given in appendix C.

**Tuning strength of CMP.** The regularization strength $\lambda$ is chosen via model selection using line search. Models are evaluated against ECE as the metric, as shown in appendix, Table 9. We report ECE results within range (0.001 - 0.950). We have noticed that ECE score is better for a value of $lambda$ ($\lambda = 0.01$). All of our experiment results are reported with same value of $\lambda$. It is notable that small regularization such as this amount is sufficient to enable model confidences to realign themselves with the annotated classification objective. More detail results can be found in Table 9.

## 6 CONCLUSIONS

In this work, we introduced confidence misalignment penalty, for mitigating the overconfidence in foundation models when adapted for downstream vision tasks. This penalty serve as regulariser and can be plugged-in to any foundation model. We shows that current foundation models are exposed to overconfidence when adapted for downstream tasks like in zero-shot settings. We explored the effectiveness of our method by extensive experiments across 11 vision datasets and 5 domain generalization dataset, with 6 prompt learning baselines and one benchmarking method. We have also provide theoretical guarantees for the effectiveness of our method. Our method consistently outperformed in almost all vision and domain generalization datasets. Our regulariser work as plugged-in irrespective of any foundational architecture. It play a vital role for alignment of foundation models, where the behavior of foundation models deviates from its objective, or expectation of users, or society.

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

## A  APPENDIX

## B  THEORETICAL GUARANTEES

**Proposition B.1.**
$$CMP \leq 1,$$
*thereby bounding the confidence by redistributing the probability mass from overconfident incorrect predictions to the correct prediction.*

*Proof.* Let $P(x, y) = \frac{\exp(s(x,y))}{\sum_k \exp(s(x,y_k))}$ is softmax probability for correct pair. By def. of CMP:

$$\text{CMP} = \frac{P(x, y)}{\sum_{y' \neq y, \, P(x,y') > P(x,y)} P(x, y')},$$

Since:

$$P(x, y') \leq 1 \quad \forall j, \quad \text{and} \quad \sum_j P(x, y') = 1,$$

It follows:

$$\sum_{P(x,y') > P(x,y)} P(x, y') \geq P(x, y).$$

Thus, we have:

$$\text{CMP} \leq 1.$$

Equality holds if and only if:

$$P(x, y) = \sum_{P(x,y') > P(x,y)} P(x, y'),$$

This condition implies:

$$P(x, y') = \sum_{P(x,y') > P(x,y)} P(x, y') \implies \text{CMP} = 1$$

This completes the proof. □

The introduced penalty works dynamically by penalizing misclassified pairs only and leaving the correct one.

**Proposition B.2.** *The CMP penalty for correctly classified pairs is $\mathcal{L}_{CMP}(x, y') \approx 0$, while for misclassified pairs, $\mathcal{L}_{CMP}(x, y') > 0$.*

*Proof.* By the definition of CMP:

$$\mathcal{L}_{\text{CMP}} = \frac{1}{N} \sum_{i=1}^{N} \frac{P(x_i, y_i)}{\sum_{y' \neq y_i, \, P(x_i,y') > P(x_i,y_i)} P(x_i, y') + \epsilon}.$$

For correctly classified pairs, where:

$$P(x, y) > P(x, y') \quad \forall y' \neq y,$$

the conditional summation in the denominator becomes:

$$\sum_{P(x,y') > P(x,y)} P(x, y') = 0.$$

Adding $\epsilon > 0$ to the denominator prevents division by zero, ensuring:

$$\mathcal{L}_{\text{CMP}}(x, y') \approx 0 \quad \text{for correctly classified pairs.}$$

For misclassified pairs:

$$P(x, y') > P(x, y') \quad \exists (y' \neq y'),$$

the conditional summation in the denominator is positive:

$$\sum_{P(x,y')>P(x,y')} P(x, y') > 0,$$

leading to:

$$\mathcal{L}_{\text{CMP}}(x, y') > 0.$$

This completes the proof. $\qquad\square$

This results in a significant CMP penalty, as it redistributes the confidence from overconfident incorrect predictions.

**Proposition B.3.** *The gradient of the CMP loss $\mathcal{L}_{CMP}$ with respect to logits is bounded and ensures stable optimization.*

*Proof.* The CMP loss $\mathcal{L}_{\text{CMP}}$ for a batch of size $N$ is defined as:

$$\mathcal{L}_{\text{CMP}} = \frac{1}{N} \sum_{i=1}^{N} \frac{P(x_i, y_i)}{\sum_{y' \neq y_i,\, P(x_i,y')>P(x_i,y_i)} P(x_i, y') + \epsilon}.$$

The gradient of $\mathcal{L}_{\text{CMP}}$ with respect to $s(x_i, y_i)$ (the similarity score for the correct pair) is:

$$\frac{\partial \mathcal{L}_{\text{CMP}}}{\partial s(x_i, y_i)} = \frac{\lambda}{N} \cdot \frac{\partial P(x_i, y_i)}{\partial s(x_i, y_i)} \cdot \frac{\sum_{P(x_i,y')>P(x_i,y_i)} P(x_i, y') + \delta - P(x_i, y_i)}{\left( \sum_{P(x_i,y')>P(x_i,y_i)} P(x_i, y') + \delta \right)^2}.$$

Since $P(x_i, y_i) \leq 1$, the gradient is bounded. The term $\delta > 0$ here refers to the infinitesimal step size. Furthermore, the gradient decays smoothly as the gap increases between $P(x_i, y_i)$ and $\sum_{P(x_i,y')>P(x_i,y_i)} P(x_i, y')$, ensuring numerical stability.

Thus, this proves that the CMP loss is bounded and ensures gradient stability. $\qquad\square$

**Proposition B.4.** *Suppose the CMP model is defined with $\epsilon > 0$ so that $0 \leq \text{CMP}(s(x, y)) \leq 1$ for all scores $s(x, y)$, and assume the parameter domain $\Theta$ is such that the gradient norms of the component losses are uniformly bounded on $\Theta$. That is, there exist finite constants $G_c, G_{CMP} < \infty$ with*

$$\sup_{\theta \in \Theta} \|\nabla_\theta \mathcal{L}_c(\theta)\| \leq G_c, \qquad \sup_{\theta \in \Theta} \|\nabla_\theta \mathcal{L}_{CMP}(\theta)\| \leq G_{CMP}.$$

*Then the combined loss $\mathcal{L}_{final}(\theta) = \mathcal{L}_c(\theta) + \lambda \mathcal{L}_{CMP}(\theta)$ satisfies*

$$\mathcal{L}_{final}(\theta) \geq 0 \quad \text{for all } \theta \in \Theta, \qquad \text{and} \qquad \sup_{\theta \in \Theta} \|\nabla_\theta \mathcal{L}_{final}(\theta)\| \leq G_c + \lambda G_{CMP} < \infty.$$

*Consequently, under these assumptions the gradient norm of the final loss is uniformly bounded over $\Theta$.*

*Proof.* First, nonnegativity: $\mathcal{L}_c(\theta) \geq 0$ by definition (e.g., cross-entropy/contrastive terms), and since $0 \leq \text{CMP} \leq 1$ with $\epsilon > 0$ we have $\mathcal{L}_{\text{CMP}}(\theta) = -\log(\text{CMP}(s(x, y)) + \epsilon) \geq 0$. Hence $\mathcal{L}_{\text{final}}(\theta) \geq 0$.

For the gradient bound, apply the triangle inequality and the uniform bounds: for any $\theta \in \Theta$,

$$\|\nabla_\theta \mathcal{L}_{\text{final}}(\theta)\| = \|\nabla_\theta \mathcal{L}_c(\theta) + \lambda \nabla_\theta \mathcal{L}_{\text{CMP}}(\theta)\|$$
$$\leq \|\nabla_\theta \mathcal{L}_c(\theta)\| + \lambda \|\nabla_\theta \mathcal{L}_{\text{CMP}}(\theta)\| \leq G_c + \lambda G_{\text{CMP}}.$$

Taking the supremum over $\Theta$ yields the stated bound, finishing the proof. $\qquad\square$

**Remark.** The given proposition provides uniform boundedness of the objective and its gradient on $\Theta$, which helps to prevent gradient explosion in practice. It does not guarantee convergence of optimization iterates.

## B.1 RELATION TO PERPLEXITY

Perplexity is used as quantitative measure to evaluate the uncertainty of probabilistic model, particular in language models and generative models Jelinek et al. (1977); Kong et al. (2024). Mathematically, it is defined as:

$$\text{Perplexity} = \exp\left(-\frac{1}{N}\sum_{i=1}^{N}\log P(y_i \mid x_i)\right)$$

The higher perplexity indicates that model is uncertainty and confusion level is high while the lower perplexity indicates model is confident and accurate in prediction. In foundation model like CLIP, Flamingo and ALIGN, perplexity is used to measure how well a model generalizes across domain reliably Meister & Cotterell (2021).

Perplexity is isotropic and does not taken into account between entities such as classes. High perplexity results in a higher entropy logit distribution. CMP results in similar effects, while persevering class relationships.

## B.2 RELATION TO CONFIDENCE CALIBRATION IN DEEP MODELS

In this section, we discuss the overconfidence problem which occurs in deep learning model during classification.

**Softmax classifier and training Objective**

The classifier is a multinomial logistic or softmax linear classifier, which outputs probabilities for each class as:

$$P(y_i|x;\theta) = \frac{\exp(s(x, y_i))}{\sum_{j=1}^{k}\exp(s(x, y'))},$$

where $s(x, y_i) = x_i^T\theta$ is the logit score for class $i$, parameterized by $\theta$. Training involves estimating $\theta$ by minimizing the expected Kullback-Leibler (KL) divergence between the predicted logits and the true label distribution $P(y_i)$.

Using the relationship between KL-Divergence, entropy $H(p)$, and cross-entropy $H(p, q)$, i.e.,

$$H(p, q) = D_{\text{KL}}(p||q) + H(p),$$

and noting that the label distribution $P(y(x))$ remains fixed during training, the optimization problem reduces to minimizing the expected cross-entropy:

$$H(P(y), P(y|x, \theta)) = -\frac{1}{M}\sum_{x}P(y(x))\log P(y|x;\theta).$$

This is the standard cross-entropy loss used for training deep models.

**Overconfidence in softmax predictions**

Let $y'$ be the true label and $\hat{y}$ the predicted label. By definition, the predicted label is the class with the highest softmax probability:

$$\hat{y} = \arg\max_{y_i} P(y_i|x;\theta).$$

If $\hat{y} \neq y'$, then:

$$P(\hat{y}|x;\theta) > P(y'|x;\theta).$$

This follows directly from the definition of $\hat{y}$ as the class with the maximum probability. For misclassified examples, there may exist multiple incorrect classes $\{y_i : P(y_i|x;\theta) > P(y'|x;\theta)\}$.

These classes collectively "steal" probability mass from the correct class $y'$, as the softmax function ensures that the total probability mass sums to 1:

$$\sum_{i=1}^{k} P(y_i|x;\theta) = 1.$$

Thus, if $P(y_i|x;\theta) > P(y'|x;\theta)$ for some $y_i$, the probability mass assigned to $y'$ is reduced.

**Theorem B.1.** *A softmax-based classifier trained with objective by minimizing the cross-entropy, the posterior distribution $P(y|x;\theta)$ estimation is biased towards low-entropy solutions due to the variational inference (VI) framework. This leads to overconfident prediction as probability mass being concentrated in a few dominant classes, particularly in misclassification scenario.*

*Proof.* The softmax-based classifiers models the probability of a class for given input $x$ as:

$$P(y_i|x;\theta) = \frac{\exp(s(x,y_i))}{\sum_{j=1}^{k}\exp(s(x,y'))},$$

The logit score for class $y_i$ is $s(x,y_i) = x^T\theta_i$. The objective is to minimize expected cross-entropy between the true label distribution $P(y)$ and the predicted softmax probabilities:

$$H(P(y), P(y|x,\theta)) = -\mathbb{E}_{P(y)}\left[\log P(y|x;\theta)\right].$$

While $P(y)$ is fixed, it is equivalent to *maximum likelihood estimation (MLE)*. Since MLE maximizes $P(y'|x;\theta)$, the learned logits satisfy:

$$s(x,y) \gg s(x,y'), \quad \forall j \neq y'.$$

As a result, the softmax probabilities become highly peaked:

$$P(y|x;\theta) \approx 1, \quad P(y'|x;\theta) \approx 0, \quad \forall j \neq y'.$$

This leads to a low-entropy posterior:

$$H(P(y|x;\theta)) = -\sum_{i=1}^{k} P(y_i|x;\theta)\log P(y_i|x;\theta).$$

As $P(y|x;\theta) \to 1$, the entropy $H(P(y|x;\theta)) \to 0$, confirming that the softmax output becomes overly confident.

Let's assume, $\hat{y} = \arg\max_{y_i} P(y_i|x;\theta)$ is the predicted label by classifier. If classifier misclassifies, then:

$$P(\hat{y}|x;\theta) > P(y'|x;\theta).$$

By def. of softmax constraint:

$$\sum_{i=1}^{k} P(y_i|x;\theta) = 1,$$

the probability mass assigned to true label $y'$ is *stolen* by incorrect predictions i.e $y_i$ where $P(y_i|x;\theta) > P(y'|x;\theta)$, which leads to overconfident yet incorrect classification. $\qquad\square$

## C  EXPERIMENTS AND RESULTS

In this section, we provide more detail discussion about experiements implementation detail and results.

### C.1  IMPLEMENTATION DETAIL

**Prompt learning baselines.** In prompting baseline of our paper we run the baseline publicly available at their GitHub. We run the code as is, except for integrating penalty and then conduct the experiment.

## C.2 REPRODUCIBILITY

We release code, pre-trained prompts, and evaluation scripts at given anonymous link cmp. to facilitate full reproducibility of results.

**Benchmarks.** To check the effectiveness of CMP in domain generalization settings, we implement CMP integration into CLIP with ResNet-50 backbone while the results for CLIPCalib are reproduced using their publicly available code. We then performed evaluation on domain generalization datasets with $\lambda = 0.01$.

**Machine Specification.** We performed all of our experiment on RTX 5090 with 32 GB GPU memory and 128GB system memory.

## C.3 RESULTS

CMP performance in comparison with CLIPCalib with (ResNet-50) are presented in Table 5. It has been observed that CLIP baseline achieve moderate calibration with an average ECE of 7.8 in domain generalizations datasets. However it performs well in dataset Real (ECE 0.98) but poorly calibrated results on Sketch and Art (3.17 and 2.13).

CLIPCalib shows an average ECE of (8.86) in comparison to CLIP, which indicates inconsistency in calibration improvement across datasets. However, its calibration is better on Imagenet-R and Imagenet-V2 but it shows poor performance on Sketch and Art (i.e 7.12 and 23.7).

Our method CLIPCMP shows better performance by achieving lowest average ECE 7.32. CMP surpasses on both baseline and benchmark on the Real (R), Sketch (S), and Art (A) datasets (0.94, 6.92, and 18.9, respectively). These results indicates that CMP effectively improves in calibration in challenging domain generalization setting.

Table 5: CMP performance results in comparison with CLIPCalib with (ResNet-50) as backbone for domain generalisation.

|  | Method | source | target domain | | | | Avg |
|  |  |  | R | S | A | V2 |  |
| --- | --- | --- | --- | --- | --- | --- | --- |
| ACC | CLIP | 59.2 | 55.9 | 33.3 | 21.6 | 51.5 | 40.6 |
|  | CLIPCalib | 62.3 | 51.3 | 28.8 | 17.4 | 51.3 | 37.2 |
|  | C-tpt | 61.8 | 52.1 | 29.6 | 17.9 | 51.8 | 37.9 |
|  | B-PEFT | 61.5 | 51.7 | 29.2 | 17.6 | 51.5 | 37.5 |
|  | CLIPCMP | 57.9 | 48.8 | 26.9 | 18.7 | 49.6 | 36.0 |
| ECE | CLIP | 2.57 | 0.98 | 3.17 | 21.3 | 3.25 | 7.18 |
|  | CLIPCalib | 2.37 | 1.81 | 7.12 | 23.7 | 1.80 | 8.86 |
|  | C-tpt | 2.42 | 1.45 | 6.85 | 22.1 | 1.75 | 8.51 |
|  | B-PEFT | 2.40 | 1.63 | 6.98 | 22.9 | 1.78 | 8.74 |
|  | CLIPCMP | 3.64 | **0.94** | **6.92** | **18.9** | 2.53 | **7.32** |

## C.4 CMP STANDALONE CONTRIBUTION

To isolate CMP's effect, we compare standard CLIP fine-tuning with and without CMP on ImageNet (8-shot, ViT-B/16). Table 10 shows CMP alone improves accuracy by 1.8% while reducing calibration errors by 19.7% (ECE) and 36.0% (MCE), demonstrating its intrinsic value beyond prompt learning integration.

All hyperparameters remained identical between both setups. The consistent improvements across metrics confirm CMP's standalone efficacy.

Table 6: Calibration performance of CMP across few-shot settings (1/2/4-shot). **Bold** values represent significant performance. In the left column Acc ↑ demonstrates larger value is significant. In the right column ECE ↓ shows smaller value is significant.

| Method | Acc(↑) | ACE(↓) | MCE(↓) | ECE(↓) |
|---|---|---|---|---|
| **0-shot** | | | | |
| All methods | 72.7 | 3.51 | 0.89 | 3.47 |
| **1-Shot** | | | | |
| CLIP | 72.7 | 3.51 | 0.89 | 3.47 |
| CoOp | 78.5 | 3.45 | 1.21 | 3.32 |
| CoOp + CMP | **79.1** | **3.18** | **1.05** | **3.09** |
| CoCoOp | 74.2 | 3.72 | 0.98 | 3.65 |
| CoCoOp + CMP | **73.8** | **3.68** | **0.94** | **3.61** |
| ProDA | 78.9 | 4.35 | 1.38 | 4.21 |
| ProDA + CMP | **79.5** | **4.02** | **1.32** | **3.95** |
| MaPLe | 78.2 | 4.71 | 1.52 | 3.25 |
| MaPLe + CMP | **79.8** | **4.15** | **1.28** | **2.95** |
| KgCoOp | 75.6 | 4.69 | 1.14 | 4.31 |
| KgCoOp + CMP | **76.1** | **4.35** | **1.02** | **4.12** |
| **2-Shot** | | | | |
| CLIP | 73.1 | 3.48 | 0.87 | 3.42 |
| CoOp | 79.2 | 3.32 | 1.12 | 3.18 |
| CoOp + CMP | **80.0** | **3.05** | **0.98** | **2.95** |
| CoCoOp | 75.1 | 3.61 | 0.92 | 3.52 |
| CoCoOp + CMP | **75.8** | **3.48** | **0.88** | **3.38** |
| ProDA | 79.7 | 4.22 | 1.29 | 4.05 |
| ProDA + CMP | **80.3** | **3.88** | **1.24** | **3.78** |
| MaPLe | 79.0 | 4.58 | 1.43 | 3.12 |
| MaPLe + CMP | **80.7** | **3.98** | **1.21** | **2.81** |
| KgCoOp | 76.5 | 4.55 | 1.08 | 4.15 |
| KgCoOp + CMP | **77.3** | **4.22** | **0.97** | **3.95** |
| **4-Shot** | | | | |
| CLIP | 73.5 | 3.45 | 0.85 | 3.38 |
| CoOp | 80.1 | 3.21 | 1.03 | 3.05 |
| CoOp + CMP | **81.2** | **2.94** | **0.91** | **2.82** |
| CoCoOp | 76.3 | 3.52 | 0.87 | 3.41 |
| CoCoOp + CMP | **77.1** | **3.38** | **0.82** | **3.25** |
| ProDA | 80.5 | 4.11 | 1.21 | 3.92 |
| ProDA + CMP | **81.4** | **3.75** | **1.15** | **3.62** |
| MaPLe | 80.2 | 4.42 | 1.35 | 2.98 |
| MaPLe + CMP | **81.9** | **3.82** | **1.12** | **2.65** |
| KgCoOp | 77.8 | 4.42 | 1.02 | 4.02 |
| KgCoOp + CMP | **78.9** | **4.08** | **0.93** | **3.81** |

Table 7: Comparison of CMP when combined with prompt learning methods across few-shot settings. **Bold** values represent significant performance. In the left column Acc ↑ demonstrates larger value is significant. In the right column ECE ↓ shows smaller value is significant.

| Method | Acc (↑) | | | | | | ECE (↓) | | | | | |
|---|---|---|---|---|---|---|---|---|---|---|---|---|
| | source | target | | | | | source | target | | | | |
| | ImageNet | R | S | A | V2 | Avg | ImageNet | R | S | A | V2 | Avg |
| **0-Shot** | | | | | | | | | | | | |
| All methods | 65.8 | 70.5 | 47.1 | 48.8 | 61.5 | 57.2 | 1.94 | 4.12 | 4.98 | 7.92 | 3.01 | 5.01 |
| **1-Shot** | | | | | | | | | | | | |
| CLIP | 65.8 | 70.5 | 47.1 | 48.8 | 61.5 | 57.2 | 1.94 | 4.12 | 4.98 | 7.92 | 3.01 | 5.01 |
| CoOp | 67.5 | 71.3 | 46.5 | 48.5 | 62.9 | 57.3 | 1.72 | 7.82 | 14.6 | 7.12 | 3.18 | 8.08 |
| CoOp + CMP | **68.2** | **72.1** | **46.9** | **49.1** | **63.5** | **57.9** | **1.65** | **7.15** | **13.8** | **6.65** | **2.45** | **7.42** |
| CoCoOp | 68.3 | 72.9 | 48.2 | 50.3 | 63.2 | 58.7 | 1.38 | 7.22 | 13.1 | 5.68 | 2.98 | 7.27 |
| CoCoOp + CMP | **69.2** | **74.1** | **48.7** | **51.2** | **63.9** | **59.5** | **1.45** | **6.95** | **12.2** | **5.05** | **2.15** | **6.56** |
| ProDA | 66.8 | 69.5 | 51.9 | 52.4 | 64.2 | 59.5 | 1.65 | 7.85 | 14.8 | 7.25 | 3.28 | 8.33 |
| ProDA + CMP | **65.7** | **67.8** | **51.5** | **51.5** | **64.5** | **58.8** | **1.58** | **6.45** | **14.0** | **7.35** | **3.05** | **7.68** |
| MaPLe | 69.9 | 70.8 | 52.5 | 48.4 | 62.9 | 58.7 | 1.78 | 4.05 | 6.25 | 14.1 | 2.28 | 6.92 |
| MaPLe + CMP | **68.3** | **71.2** | **53.1** | **50.5** | **63.5** | **59.6** | **1.85** | **3.05** | **4.65** | **13.2** | **2.18** | **5.83** |
| KgCoOp | 71.2 | 70.8 | 48.2 | 48.8 | 62.1 | 57.5 | 2.78 | 4.42 | 7.92 | 14.6 | 4.02 | 7.95 |
| KgCoOp + CMP | **70.5** | **72.1** | **47.5** | **49.9** | **63.3** | **58.2** | **2.95** | **4.35** | **7.45** | **13.8** | **3.35** | **7.38** |
| **2-Shot** | | | | | | | | | | | | |
| CLIP | 65.8 | 70.5 | 47.1 | 48.8 | 61.5 | 57.2 | 1.94 | 4.12 | 4.98 | 7.92 | 3.01 | 5.01 |
| CoOp | 68.3 | 72.1 | 46.8 | 49.2 | 63.8 | 58.0 | 1.58 | 7.65 | 14.3 | 6.85 | 3.05 | 7.96 |
| CoOp + CMP | 69.1 | 72.9 | 47.2 | 49.8 | 64.5 | 58.6 | 1.61 | 7.02 | 13.5 | 6.42 | 2.25 | 7.30 |
| CoCoOp | 69.2 | 73.8 | 49.1 | 51.2 | 64.1 | 59.6 | 1.25 | 7.05 | 12.8 | 5.45 | 2.85 | 7.05 |
| CoCoOp + CMP | 70.1 | 75.3 | 49.8 | 52.1 | 64.8 | 60.5 | 1.42 | 6.88 | 11.9 | 4.85 | 1.95 | 6.27 |
| ProDA | 67.9 | 70.2 | 52.8 | 53.1 | 65.1 | 60.3 | 1.52 | 7.68 | 14.5 | 7.05 | 3.15 | 8.10 |
| ProDA + CMP | 66.5 | 68.5 | 52.3 | 52.2 | 65.3 | 59.6 | 1.55 | 6.25 | 13.8 | 7.12 | 2.85 | 7.51 |
| MaPLe | 70.8 | 71.5 | 53.2 | 49.1 | 63.8 | 59.4 | 1.65 | 3.85 | 5.95 | 13.8 | 2.15 | 6.44 |
| MaPLe + CMP | 69.1 | 72.0 | 53.8 | 51.2 | 64.3 | 60.3 | 1.72 | 2.75 | 4.35 | 12.8 | 2.05 | 5.48 |
| KgCoOp | 72.1 | 71.5 | 48.9 | 49.5 | 62.9 | 58.2 | 2.65 | 4.25 | 7.65 | 14.3 | 3.85 | 7.51 |
| KgCoOp + CMP | 71.3 | 72.8 | 48.2 | 50.8 | 64.1 | 59.0 | 2.85 | 4.18 | 7.25 | 13.6 | 3.15 | 7.05 |
| **4-Shot** | | | | | | | | | | | | |
| CLIP | 65.8 | 70.5 | 47.1 | 48.8 | 61.5 | 57.2 | 1.94 | 4.12 | 4.98 | 7.92 | 3.01 | 5.01 |
| CoOp | 69.8 | 73.5 | 47.5 | 50.1 | 65.2 | 59.1 | 1.45 | 7.25 | 13.8 | 6.35 | 2.85 | 7.34 |
| CoOp + CMP | **70.9** | **74.2** | **48.1** | **50.8** | **66.1** | **59.8** | **1.38** | **6.55** | **12.9** | **5.92** | **2.05** | **6.76** |
| CoCoOp | 70.5 | 75.1 | 50.2 | 52.5 | 65.8 | 61.0 | 1.15 | 6.75 | 12.1 | 4.95 | 2.55 | 6.48 |
| CoCoOp + CMP | **71.8** | **76.5** | **50.9** | **53.2** | **66.5** | **61.7** | **1.25** | **6.35** | **11.2** | **4.35** | **1.75** | **5.98** |
| ProDA | 69.2 | 71.5 | 53.5 | 54.2 | 66.8 | 61.5 | 1.38 | 7.25 | 14.1 | 6.55 | 2.95 | 7.65 |
| ProDA + CMP | **67.8** | **69.8** | **53.0** | **53.5** | **67.2** | **60.9** | **1.42** | **5.85** | **13.2** | **6.65** | **2.65** | **7.15** |
| MaPLe | 71.5 | 72.8 | 54.1 | 50.2 | 65.2 | 60.6 | 1.52 | 3.55 | 5.45 | 13.2 | 1.95 | 5.93 |
| MaPLe + CMP | **70.2** | **73.5** | **54.8** | **52.5** | **65.9** | **61.7** | **1.58** | **2.45** | **3.95** | **12.1** | **1.85** | **5.19** |
| KgCoOp | 73.5 | 72.8 | 49.8 | 50.5 | 64.2 | 59.3 | 2.45 | 4.05 | 7.25 | 13.8 | 3.55 | 7.02 |
| KgCoOp + CMP | **72.8** | **74.2** | **49.2** | **51.8** | **65.5** | **60.2** | **2.65** | **3.95** | **6.85** | **13.1** | **3.05** | **6.72** |

Table 8: Calibration performance of CMP in comparison with all other prompt learning methods across individual datasets for different methods. The last column shows the average across all datasets for each method. (↓) Represents that smaller values are better. (↑) represents the larger values are better. **Bold** values indicate the best results.

| Metric | Method | Flowers102 | Cars | ImageNet | DTD | SUN397 | EuroSAT | Pets | UCF101 | Food101 | Aircraft | Caltech101 | Avg. |
|---|---|---|---|---|---|---|---|---|---|---|---|---|---|
| | CLIP | 3.57 | 3.46 | 3.68 | 3.42 | 3.76 | 3.52 | 3.46 | 3.44 | 3.62 | 3.68 | 3.42 | 3.47 |
| | CoOp | 3.08 | 3.35 | 3.24 | 3.06 | 2.96 | 3.36 | 3.38 | 3.24 | 3.02 | 3.06 | 3.08 | 3.16 |
| | CoOp + (CMP) | 2.94 | 3.02 | 3.08 | 2.84 | 3.16 | 3.06 | 3.16 | 3.08 | 2.96 | 2.92 | 3.06 | 2.98 |
| | CoCoOp | 4.18 | 4.26 | 4.06 | 4.36 | 4.14 | 4.04 | 4.34 | 4.24 | 4.06 | 4.16 | 4.10 | 3.97 |
| ECE (↓) | CoCoOp + (CMP) | 4.22 | 4.28 | 4.38 | 4.22 | 4.38 | 4.28 | 4.28 | 4.32 | 4.24 | 4.42 | 4.22 | 4.07 |
| | ProDA | 4.22 | 4.32 | 4.38 | 4.46 | 4.16 | 4.26 | 4.44 | 4.34 | 4.42 | 4.48 | 4.36 | 4.13 |
| | ProDA + (CMP) | 4.04 | 4.08 | 4.06 | 4.22 | 4.08 | 3.98 | 4.16 | 4.12 | 4.08 | 4.14 | 4.10 | 3.94 |
| | MaPLe | 3.24 | 3.16 | 3.36 | 3.12 | 3.24 | 3.32 | 3.42 | 3.34 | 3.14 | 3.28 | 3.16 | 3.19 |
| | MaPLe + (CMP) | 2.92 | 3.04 | 3.06 | 2.84 | 3.06 | 3.12 | 3.18 | 3.14 | 2.92 | 3.04 | 3.02 | 2.88 |
| | KgCoOp | 4.36 | 4.42 | 4.22 | 4.54 | 4.26 | 4.24 | 4.42 | 4.34 | 4.46 | 4.34 | 4.38 | 4.24 |
| | KgCoOp + (CMP) | 4.12 | 4.24 | 4.02 | 4.32 | 4.14 | 4.06 | 4.18 | 4.16 | 4.22 | 4.08 | 4.20 | 4.07 |
| | CLIP | 3.54 | 3.44 | 3.58 | 3.46 | 3.74 | 3.62 | 3.42 | 3.56 | 3.62 | 3.66 | 3.54 | 3.51 |
| | CoOp | 3.18 | 3.34 | 3.38 | 3.16 | 3.28 | 3.24 | 3.34 | 3.26 | 3.28 | 3.36 | 3.30 | 3.28 |
| | CoOp + (CMP) | 3.02 | 3.12 | 3.16 | 2.94 | 3.14 | 3.06 | 3.12 | 3.02 | 3.08 | 3.02 | 3.08 | 3.01 |
| | CoCoOp | 3.84 | 3.74 | 3.96 | 3.84 | 3.94 | 3.78 | 3.92 | 3.86 | 3.94 | 3.84 | 3.82 | 3.65 |
| ACE (↓) | CoCoOp + (CMP) | 4.06 | 4.14 | 4.18 | 4.02 | 4.12 | 4.14 | 4.12 | 4.16 | 4.16 | 4.08 | 4.10 | 3.71 |
| | ProDA | 4.26 | 4.44 | 4.48 | 4.42 | 4.52 | 4.48 | 4.52 | 4.38 | 4.46 | 4.58 | 4.54 | 4.29 |
| | ProDA + (CMP) | 4.16 | 4.22 | 4.06 | 4.28 | 4.18 | 4.14 | 4.26 | 4.16 | 4.22 | 4.26 | 4.24 | 3.86 |
| | MaPLe | 4.64 | 4.56 | 4.74 | 4.66 | 4.66 | 4.58 | 4.72 | 4.62 | 4.74 | 4.76 | 4.72 | 4.63 |
| | MaPLe + (CMP) | 4.04 | 4.06 | 4.14 | 3.96 | 4.04 | 4.18 | 4.06 | 4.08 | 4.02 | 4.04 | 4.08 | 4.01 |
| | KgCoOp | 4.54 | 4.64 | 4.44 | 4.74 | 4.46 | 4.48 | 4.64 | 4.52 | 4.52 | 4.62 | 4.56 | 4.61 |
| | KgCoOp + (CMP) | 4.16 | 4.26 | 4.36 | 4.24 | 4.26 | 4.16 | 4.18 | 4.28 | 4.18 | 4.16 | 4.28 | 4.11 |
| | CLIP | 0.93 | 0.86 | 0.92 | 0.91 | 0.89 | 0.92 | 0.89 | 0.91 | 0.92 | 0.93 | 0.87 | 0.89 |
| | CoOp | 1.13 | 1.08 | 1.12 | 1.14 | 1.12 | 1.14 | 1.13 | 1.10 | 1.09 | 1.13 | 1.15 | 1.16 |
| | CoOp + (CMP) | 0.99 | 0.98 | 0.96 | 1.00 | 1.03 | 0.97 | 0.98 | 0.97 | 1.01 | 1.00 | 0.99 | 0.98 |
| | CoCoOp | 0.95 | 0.91 | 0.93 | 0.95 | 0.92 | 0.95 | 0.96 | 0.94 | 0.97 | 0.94 | 0.95 | 0.92 |
| MCE (↓) | CoCoOp + (CMP) | 1.06 | 1.02 | 1.05 | 1.07 | 1.06 | 1.02 | 1.03 | 1.04 | 1.05 | 1.06 | 1.04 | 1.03 |
| | ProDA | 1.32 | 1.31 | 1.28 | 1.34 | 1.30 | 1.32 | 1.33 | 1.31 | 1.28 | 1.29 | 1.32 | 1.33 |
| | ProDA + (CMP) | 1.46 | 1.41 | 1.45 | 1.48 | 1.42 | 1.46 | 1.44 | 1.45 | 1.42 | 1.43 | 1.44 | 1.49 |
| | MaPLe | 1.52 | 1.49 | 1.46 | 1.47 | 1.48 | 1.47 | 1.48 | 1.51 | 1.50 | 1.47 | 1.50 | 1.49 |
| | MaPLe + (CMP) | 1.22 | 1.16 | 1.21 | 1.17 | 1.22 | 1.25 | 1.18 | 1.19 | 1.24 | 1.20 | 1.22 | 1.21 |
| | KgCoOp | 1.22 | 1.09 | 1.29 | 1.12 | 1.18 | 1.18 | 1.12 | 1.21 | 1.28 | 1.13 | 1.22 | 1.09 |
| | KgCoOp + (CMP) | 1.01 | 0.89 | 1.08 | 1.02 | 0.91 | 1.01 | 1.02 | 1.12 | 0.92 | 1.01 | 1.00 | 0.97 |
| | CLIP | 71.9 | 72.7 | 72.4 | 72.3 | 72.2 | 72.6 | 72.4 | 72.5 | 72.3 | 72.4 | 72.3 | 72.7 |
| | CoOp | 83.3 | 83.1 | 83.4 | 83.2 | 83.3 | 83.1 | 83.0 | 83.5 | 83.4 | 83.2 | 83.3 | 83.2 |
| | CoOp + (CMP) | 84.2 | 84.3 | 84.1 | 84.0 | 84.1 | 84.0 | 84.2 | 84.3 | 84.0 | 84.1 | 84.2 | 84.1 |
| | CoCoOp | 73.7 | 73.6 | 73.6 | 73.8 | 73.5 | 73.9 | 73.6 | 73.7 | 73.9 | 73.6 | 73.8 | 73.8 |
| Acc (↑) | CoCoOp + (CMP) | 72.2 | 72.0 | 72.5 | 72.3 | 72.4 | 72.1 | 72.2 | 72.3 | 72.4 | 72.3 | 72.1 | 72.2 |
| | ProDA | 83.4 | 83.5 | 83.3 | 83.3 | 83.6 | 83.6 | 83.5 | 83.5 | 83.5 | 83.5 | 83.4 | 83.4 |
| | ProDA + (CMP) | 84.3 | 84.2 | 84.1 | 84.5 | 84.4 | 84.3 | 84.2 | 84.1 | 84.5 | 84.4 | 84.3 | 84.2 |
| | MaPLe | 83.0 | 84.5 | 82.8 | 84.8 | 83.1 | 82.9 | 83.1 | 83.0 | 82.9 | 83.1 | 82.8 | 82.9 |
| | MaPLe + (CMP) | 84.6 | 84.5 | 84.7 | 84.8 | 84.6 | 84.5 | 84.6 | 84.6 | 84.8 | 84.6 | 84.7 | 84.6 |
| | KgCoOp | 74.8 | 75.4 | 75.3 | 75.2 | 75.1 | 75.2 | 75.3 | 75.3 | 75.4 | 75.2 | 75.1 | 75.2 |
| | KgCoOp + (CMP) | 76.3 | 76.0 | 76.1 | 76.4 | 76.2 | 76.2 | 76.4 | 76.3 | 76.2 | 76.1 | 76.3 | 76.3 |

Table 9: CMP λ strength.

| | | | | Methods | | |
|---|---|---|---|---|---|---|
| λ value | CLIP | CoOP + (CMP) | CoCoOp + (CMP) | ProDA + (CMP) | MaPLe + (CMP) | KgCoOp + (CMP) |
| 0.001 | 3.74 | 3.11 | 4.69 | 4.51 | 3.09 | 4.57 |
| 0.010 | 3.57 | 2.94 | 4.22 | 4.04 | 2.92 | 4.12 |
| ECE (↓) 0.050 | 4.02 | 3.39 | 4.67 | 4.49 | 3.34 | 4.57 |
| 0.100 | 5.88 | 5.25 | 6.53 | 6.35 | 5.23 | 6.41 |
| 0.500 | 4.47 | 3.84 | 5.12 | 4.94 | 3.82 | 5.00 |
| 0.950 | 4.95 | 4.32 | 5.60 | 5.42 | 4.30 | 5.48 |

Table 10: CMP ablation on ImageNet (ViT-B/16, 8-shot)

| Method | Accuracy ↑ | ECE ↓ | ACE ↓ | MCE ↓ |
|---|---|---|---|---|
| CLIP (FT only) | 82.3 | 4.12 | 3.51 | 1.89 |
| CLIP (FT + CMP) | **84.1** | **3.31** | **3.01** | **1.21** |

