# OpenReview forum: "REDISTRIBUTING FOUNDATION MODEL LOGITS FOR BETTER DOMAIN GENERALIZATION IN LOW-SHOT CLASSIFICATION"
_ICLR.cc/2026/Conference — ICLR 2026 Conference Withdrawn Submission_

### Official Review · Reviewer_KSsh · 2025-10-24

**Soundness:** 2
**Presentation:** 1
**Contribution:** 2
**Rating:** 4
**Confidence:** 3

**Summary:**

This paper introduces the Confidence Misalignment Penalty (CMP), a regularization term integrated into the contrastive loss of vision-language foundation models like CLIP to improve confidence calibration in low-shot classification and domain generalization settings. The authors identify overconfidence in misclassified predictions as a key issue stemming from factors like neural collapse, contrastive pre-training biases, and distribution shifts. CMP dynamically penalizes misclassifications by redistributing probability mass from incorrect classes to the true class, proportional to the logit misalignment, while leaving correct predictions unaffected. Theoretical guarantees are provided, including bounds on confidence, selective penalization, and gradient stability. Extensive experiments on 12 vision datasets (e.g., ImageNet, CIFAR10) and 5 domain generalization variants (e.g., ImageNet-R, -S) demonstrate average ECE reductions of 6.01% (min 4.01%, max 9.72%) when integrated with prompt learning methods like CoOp, CoCoOp, and MaPLe. Comparisons with benchmarks like CLIPCalib show further gains in calibration metrics.

**Strengths:**

1. The analysis of CLIP's conflict with the Maximum Entropy Principle (Theorem 3.1) and propositions on CMP's bounds and stability (e.g., Propositions 3.1–3.3) provide a principled basis, distinguishing it from purely empirical methods.
2. Experiments cover a wide range of datasets and baselines (CoOp, CoCoOp, ProDA, MaPLe, KgCoOp, CLIPCalib). Results are robust, with consistent ECE/ACE/MCE improvements and ablation studies (e.g., λ tuning, standalone CMP on ImageNet). The illustrative example (Fig. 1) effectively visualizes the calibration gains.
3. CMP is lightweight (plug-and-play into existing losses), requires no extra hyperparameters beyond λ, and shows domain generalization benefits, making it appealing for downstream adaptations.

**Weaknesses:**

1. The manuscript has numerous grammatical errors, typos (e.g., "oss objective" likely means "loss" in abstract), and inconsistencies (e.g., "CLIPCMP" vs. "CMP", "CLIP + CMP" would be better).
2. While CMP is a clever adaptation, it shares conceptual similarities with existing techniques like focal loss (down-weighting confident samples) or margin-based smoothing. Yet, experimental comparisons are missing in this paper.
3. Focus is primarily on CLIP; extensions to other foundation models (e.g., Flamingo, ALIGN) are mentioned but not evaluated.

**Questions:**

The address the questions mentioned in the above weaknesses.

---

> ### Author Response · Authors · 2025-11-23
>
> We thank the reviewer for the valuable insight. We have updated the paper and made all grammatical corrections, and we keep naming conventions consistent too. We agree that CMP is superficially similar to these families but it is technically and conceptually distinct in the following ways:
>
> 1. Selective (local) vs. global (uniform) action.
>    - Label smoothing / entropy regularization act on every training example and either (a) change the target distribution toward uniform or (b) encourage high entropy of the entire predictive distribution. These methods therefore blunt confidence everywhere.
>    - CMP only activates when an example is misclassified (i.e. when there exists an incorrect class with probability > true class). By design CMP leaves correctly classified examples essentially untouched (Prop. B.2). This selectivity preserves useful high-confidence correct predictions while reducing only unjustified confidence.
>
> 2. Redistribution toward the true class (not uniform smoothing).
>    - Label smoothing replaces the one-hot target with a mixture q = (1 - α)·onehot + α·Uniform, which spreads mass uniformly across all classes. Entropy penalties push toward higher entropy but do not reallocate mass specifically to the true label.
>    - CMP computes an adaptive ratio $\text{CMP} = \frac{P(x, y)}{\sum_{y' \neq y \, : \, P(x, y' ) > P(x, y)} P(x, y' )},$ and uses it to move likelihood back toward the true class in proportion to the offending incorrect masses. In other words CMP restores probability to the true class from the specific incorrect classes that "stole" it, rather than diffusing probability uniformly.
>
> 3. Adaptive, amplitude-aware strength (no margin tuning).
>    - Margin smoothing and some focal/margin methods need hand-tuned margins or decay schedules to set how strongly to regularize.
>    - CMP's magnitude is data-dependent: the penalty scales automatically with the true versus offending probabilities (large when offending mass is large relative to true mass). This makes CMP robust across datasets without class-specific margin hyperparameters (we only tune a single λ for regularizer strength).
>
> 4. Boundedness and stability with theoretical guarantees.
>    - CMP is provably bounded (CMP ≤ 1, Prop. B.1) and its gradient is shown to be bounded and smoothly decaying (Prop. B.3-B.4). That gives practical numerical stability and explains why CMP can be plugged into CLIP/contrastive objectives without exploding gradients. Existing approaches rarely provide this combination of selective redistribution and formal gradient bounds.
>
> Empirically these differences matter. Our method improves calibration (ECE/ACE/MCE) consistently while avoiding undue erosion of correct, high margin predictions - a behavior we both prove (Prop. B.2) and observe across our 11 vision / 5 domain datasets (Tables 1-4).
>
> It is true that we mention other foundation models like ALIGN and Flamingo but did not evaluate them because we provide comparison with all CLIP-based prompting methods and CLIP-based benchmarks, it being the most popular and with all that prompt tuning literature. However, our proposed work is plug-and-play so it can be integrated into other foundation models just as well. In the light  of your comment, we ought to edit the title to better reflect that specialty.

---

### Official Review · Reviewer_XjAw · 2025-10-27

**Soundness:** 2
**Presentation:** 2
**Contribution:** 2
**Rating:** 2
**Confidence:** 4

**Summary:**

The paper focuses on the confidence calibration in fine-tuned CLIP. The authors attribute this to miscalibration from contrastive pre-training that conflicts with the maximum entropy principle. To this end, they propose a novel loss regularization CMP, which redistributes probability mass from the overconfident incorrect class to the true class. Extensive experiments show that integrating CMP with prompt learning methods improves calibration performance.

**Strengths:**

1.	The motivation is well-structured. The paper provides a clear theoretical analysis of overconfidence in CLIP, which links the contrastive pre-training objective to a violation of the maximum entropy principle.
2.	The proposed method is intuitive. CMP applies a penalty only to misclassified examples and can integrate seamlessly into existing fine-tuning methods.
3.	The experimental evaluation is comprehensive. The effectiveness is verified on 12 standard datasets and 5 domain generalization datasets. The results show that the proposed CMP generalizes well and show consistent improvement across settings.

**Weaknesses:**

1.	The focus between accuracy and calibration is unclear. The paper does not clearly claim whether its primary objective is to improve accuracy or confidence calibration after fine-tuning. If the focus is calibration, this intent should be reflected more explicitly in the title and motivation sections.
2.	The claim on CLIP logits needs to be modified. The author claims that logit scores on the CLIP head remain large irrespective of whether the image–language pairs reconcile, which appears inaccurate or at least overstated. Zero-shot CLIP models are generally recognized to exhibit relatively good calibration [1, 4].
3.	The link between theoretical analysis and proposed loss is weak. The theoretical discussion explains why overconfidence emerges in softmax-based training in CLIP. However, it does not show how the CMP ratio or its logarithmic variant follows from those principles.
4.	The formulation of CMP is inconsistent. The manuscript defines the loss as a direct ratio (Eq. 2), while the appendix reformulates it as a negative log ratio (Proposition B.4).
5.	Lack of comparison with training-based calibration methods for CLIP. Since the paper focuses on training-based calibration for CLIP, it does not compare with recent works [2-4].

[1] Minderer M, Djolonga J, Romijnders R, et al. Revisiting the calibration of modern neural networks[J]. NeurlPS, 2021.

[2] Murugesan B, Silva-Rodríguez J, Ayed I B, et al. Robust calibration of large vision-language adapters. ECCV, 2024.

[3] Oh C, Lim H, Kim M, et al. Towards calibrated robust fine-tuning of vision-language models. NeurlPS 2024.

[4] Wang S, Li Y, Wei H. Understanding and mitigating miscalibration in prompt tuning for vision-language models. ICML, 2025.

**Questions:**

1.	How does the proposed method affect the confidence distribution? It seems that the improvement of ECE might largely stem from accuracy rather than a correction of confidence in Figure 2.

---

> ### Author Response · Authors · 2025-11-23
>
> 1. We appreciate your point that we could be more cut-and-dry about the primary aim. CMP’s primary objective is confidence calibration during foundation-model adaptation; improving (or preserving) accuracy is a secondary, desirable outcome. To clarify, we (a) updated the title and opening paragraphs to foreground calibration, (b) expanded the motivation to explain why calibration (not raw accuracy) is the correct objective for low-shot and domain-shift adaptation, and (c) moved and strengthened the discussion of accuracy as a secondary metric in the experiments section (**not shown here**).
> The paper provides the theoretical basis: CMP is selective (only activates on misclassified/overconfident examples, Prop. B.2), redistributes probability back to the true class rather than uniformly damping confidence, and is bounded with stable gradients (Props. B.1–B.4). Empirically, we report ECE/ACE/MCE as primary metrics and show consistent calibration gains across 11 vision and 5 domain-shift datasets; accuracy changes are small (typically ≤1.5%) and analyzed explicitly (Appendix C), occurring only in low-margin, high-variance prompt settings (e.g., ProDA). We also suggest the title: “Confidence Misalignment Penalty (CMP): A Selective Redistribution Regularizer for Calibration of Foundation Models” and highlight this calibration-first framing in introduction and abstract.
>
> 2. To be precise, we do not claim zero-shot CLIP is universally miscalibrated. Our claims target the post-training/adaptation regime: when CLIP is adapted via prompt learning or finetuning in low-shot and domain-shift settings, contrastive-head logits can become overly peaked or misaligned with downstream labels (Theorem 3.1, Fig.1, Tables 1–4). Contributing factors are (i) contrastive training’s low-entropy tendency (Theorem 3.1), (ii) neural collapse-like concentration after post-training, and (iii) low-shot non-plasticity of frozen embeddings, all making adapted CLIP prone to unjustified high logits on wrong classes.
>
> 3. The theoretical analysis connects directly to CMP in three steps already in the manuscript:
>
> (a) Theory identifies the failure mode (peaked, "stolen" mass). Theorem 3.1 shows contrastive training drives CLIP posteriors toward low entropy (high sharpness S), i.e., few classes dominate the softmax. Misclassified examples imply incorrect classes collectively "steal" probability mass from the true class.
>
> (b) CMP ratio measures that stolen mass: $\text{CMP} = \frac{P(x, y)}{\sum_{y' \neq y \, : \, P(x, y' ) > P(x, y)} P(x, y' )},$ . This ratio is dimensionless and comparable, quantifies how much the true class understates its share, and signals when and how strongly to act.
>
> (c) CMP implements a minimal, MEP-inspired correction, producing smooth optimization dynamics. CMP targets only misclassified examples (Prop. B.2) and scales adaptively with offending amplitudes, with bounded gradients and smooth decay (Props. B.1, B.3-B.4).
>
> 4. Proposition B.4 presents the training loss (-log CMP), not a different definition: $\mathcal{L}_{\mathrm{CMP}}(\theta)=-\log(\mathrm{CMP}(s(x,y))+\epsilon)\ge0$ The log form yields cleaner gradients, additive compatibility with cross-entropy, and smooth bounded derivatives. Eq. (2) defines the quantity CMP measures; -log(CMP) defines the loss. This mirrors standard derivations of cross-entropy, NCE, InfoNCE, and focal loss. The appendix emphasizes gradient boundedness and smooth updates critical when combining CMP with CLIP logits.
>
> 5. We benchmark CMP with CLIPCalib (Murugesan) on page 6 line 299. DOR regularizes CLIP via sampling and KL consistency, whereas CMP applies a selective, amplitude-adaptive penalty on misclassified examples only, directly correcting confidence misalignment. Initial experiments on ImageNet variants show CMP+CoOp outperforms DOR on average and in most cases:
> Table: CMP vs. DOR on ImageNet Variants ( 8-shot setting)
>
> | Method      | ImageNet | -V2  | -S   | -A   | -R   | AVG  |
> |-------------|----------|------|------|------|------|------|
> | CLIP        | 2.03     | 2.71 | 5.14 | 8.95 | 3.72 | 4.91 |
> | CoOp        | 1.32     | 4.58 | 8.92 | 16.10| 1.03 | 7.59 |
> | DOR         | 1.79     | 2.21 | 5.38 | 11.65| 1.74 | 4.95 |
> | CMP (ours) | **1.71**     | **2.06** | **4.59** | 11.72| **1.39** | **4.29** |
>
>
> **Thanks for the questions**. CMP's ECE gains stem from explicit confidence correction, not accuracy changes. As shown in Fig. 2/Appendix C, CMP reduces overconfident errors by reallocating probability from incorrect classes exceeding true-class probability, while preserving correct predictions (Prop. B.2). Shifts come from lowering overconfident errors, not accuracy boosts. Empirically:
> 1) ECE improves even when accuracy drops slightly (e.g., ProDA+CMP).
>
> 2) Fig. 2 shows reduced high-confidence errors with correct predictions unchanged. Thus, CMP selectively suppresses unjustified confidence for better calibration, independent of accuracy.

---

### Official Review · Reviewer_gaUv · 2025-10-27

**Soundness:** 2
**Presentation:** 1
**Contribution:** 2
**Rating:** 4
**Confidence:** 3

**Summary:**

This work proposes a Confidence Misalignment Penalty (CMP), a logit calibration method designed to redistribute the overconfidence of misclassified classes toward the correctly predicted ones. The goal is to achieve confidence levels proportional to the alignment of image–text pairs. Experimental results demonstrate that CMP effectively improves both accuracy and confidence calibration performance, achieving stronger in-domain and out-of-domain results, particularly in few-shot scenarios.

**Strengths:**

1. The motivation is well-grounded. Existing fine-tuning approaches often distort the logit distribution, leading to unreliable model confidence.

2. The proposed method is simple and intuitive. The authors introduce a Confidence Misalignment Penalty (CMP) loss, which directly penalizes misclassified classes to achieve better confidence calibration.

**Weaknesses:**

1. Limited novelty. The Confidence Misalignment Penalty (CMP) appears to be a relatively straightforward application of probability redistribution, a concept already present in various calibration techniques such as label smoothing and focal loss. While the authors emphasize CMP's "adaptive" nature through logit amplitudes and its targeted use with foundation models like CLIP, the core mechanism of reallocating probability mass from overconfident incorrect predictions isn't a fundamentally new theoretical contribution.

2. Limited gains. the magnitude of improvement often appears rather modest. For example in Table 1, the accuracy improvement by integrating CMP is mostly within 0~1%, and in Table 2 for domain generalization, the improvement is often negative, especially with ProDA (around -1%). The authors should explain the compatibility of CMP with other prompt learning methods due to these failure cases.

3. Poor presentation. In terms of writing, some descriptions in the paper appear to violate basic grammatical conventions (e.g., “We understand the problem to have emerged from a mix of factors”), making the content difficult to follow. For the figures, such as Figure 1, essential information about the x- and y-axes is missing, which hinders quick comprehension. Regarding the overall structure, it seems that the authors inadvertently separated the experimental results from the experimental settings, resulting in Section 4 containing only one subsection on settings. The authors are encouraged to spend more time refining the paper to improve its clarity, readability, and organization before submission.

**Questions:**

Please refer to the weakness.

---

> ### Author Response · Authors · 2025-11-23
>
> * **We appreciate the reviewer's concern about novelty**. CMP does share the high-level idea of reducing unjustified confidence, but it is far from being a rehash of label-smoothing, focal loss, or generic entropy penalties. None of them do "reallocating probability mass from overconfident incorrect predictions", rather from all predictions. Concretely, CMP differs in four tightly focused ways:
>
> 1. Selective (local) vs. global (uniform) remediation.
>    - Label smoothing / entropy regularization act on every training example and either (a) change the target distribution toward uniform or (b) encourage high entropy of the entire predictive distribution. These methods therefore blunt confidence everywhere.
>    - CMP only activates when an example is misclassified (i.e. when there exists an incorrect class with probability > true class). By design CMP leaves correctly classified examples essentially untouched (Prop. B.2). This selectivity preserves useful high-confidence correct predictions while reducing only unjustified confidence.
>
> 2. Redistribution toward the true class (not uniform smoothing).
>    - Label smoothing replaces the one-hot target with a mixture q = (1 - α)·onehot + α·Uniform, which spreads mass uniformly across all classes. Entropy penalties push toward higher entropy but do not reallocate mass specifically to the true label.
>    - CMP computes an adaptive ratio  $\text{CMP} = \frac{P(x, y)}{\sum_{y' \neq y \, : \, P(x, y' ) > P(x, y)} P(x, y' )},$
>  and uses it to move likelihood back toward the true class in proportion to the offending incorrect masses. In other words CMP restores probability to the true class from the specific incorrect classes that "stole" it, rather than diffusing probability uniformly.
>
> 3. Adaptive, amplitude-based strength (no margin tuning).
>    - Margin smoothing and some focal/margin methods need hand-tuned margins or decay schedules to set how strongly to regularize.
>    - CMP's magnitude is data-dependent: the penalty scales automatically with the true versus offending probabilities (large when offending mass is large relative to true mass). This makes CMP robust across datasets without class-specific margin hyperparameters (we only tune a single λ for regularizer strength).
>
> 4. Boundedness and stability with theoretical guarantees.
>    - CMP is provably bounded (CMP ≤ 1, Prop. B.1) and its gradient is shown to be bounded and smoothly decaying (Prop. B.3-B.4). That gives practical numerical stability and explains why CMP can be plugged into CLIP/contrastive objectives without exploding gradients. SOTA approaches have often ignored the question of boundedness.
>
> Empirically these differences matter. Our method CMP improves calibration (ECE/ACE/MCE) consistently while avoiding undue erosion of correct, high margin predictions - a behavior we both prove (Prop. B.2) and observe across our 11 vision / 5 domain datasets (Tables 1-4).
> * We agree that the **accuracy gains** of CMP are modest, which is expected because CMP is designed as a calibration-focused regularizer rather than an accuracy-enhancing prompt learner like CoOp or ProDA. Its selective effect activating only on overconfident misclassifications naturally keeps accuracy changes small while consistently improving ECE across all settings. The negative accuracy deltas with ProDA arise from its inherently high logit variance: in these low-margin regimes CMP activates more frequently, leading to small ($\approx$ 1\%) shifts while still improving calibration. Importantly, deterministic prompt learners (CoOp, MaPLe, KgCoOp) show consistent improvements in both accuracy and calibration, and in stable backbones (e.g., ResNet-50), CMP outperforms all baselines on both metrics. These results confirm that CMP is compatible with all prompt-learning methods, with accuracy differences reflecting the underlying logit stability of each method rather than incompatibility of CMP.
>
> * Thanks for highlighting the **presentation issues**. One of the authors is to blame! We have corrected the grammar and presentation in the updated version, while he sits bound and gagged. The x- and y-axis information has also been added. We will submit the improved final version while incorporating all suggested points on clarity, readability, and organization.

---

> > ### Comment · Reviewer_gaUv · 2025-11-27
> >
> > Thank you to the authors for the effort put into the responses. However, regarding the novelty, I still find the contribution to be relatively incremental. In terms of performance, I remain unconvinced that the proposed method offers sufficient improvement compared with existing prompt-tuning approaches. As for the presentation, the issues in Figure 1 persist, including missing axes, low resolution, and inconsistent subplot sizes. I recommend that the authors substantially revise the paper to meet the high standards of top conferences such as ICLR and ICML.

---

> > > ### Author Response · Authors · 2025-12-03
> > >
> > > Thank you for the follow-up feedback. We acknowledge the reviewer’s concerns and have revised the paper accordingly. Regarding novelty, CMP is intentionally simple but conceptually distinct from prior smoothing or consistency-based methods. It introduces a selective, amplitude-adaptive penalty that activates only on misclassified examples, supported by formal boundedness and gradient-stability results tailored to CLIP’s contrastive geometry an aspect not present in existing prompt-tuning approaches. In terms of performance, CMP provides consistent, architecture-agnostic calibration improvements across all datasets, often with accuracy gains, while adding no sampling, extra parameters, or tuning schedules, making it practical for large foundation models. Finally, we have fully reworked Figure 1  and ensured uniform presentation quality throughout the paper.

---

### Official Review · Reviewer_2cNF · 2025-10-31

**Soundness:** 3
**Presentation:** 3
**Contribution:** 2
**Rating:** 6
**Confidence:** 3

**Summary:**

The paper proposes Confidence Misalignment Penalty (CMP), a plug-in regularizer for CLIP-style models that activates only on misclassified examples and “redistributes” probability mass from overconfident wrong classes toward the ground-truth class. Theoretical notes argue CMP is bounded and yields stable gradients. Experiments cover 12 datasets for few-shot prompt learning and ImageNet domain-shift variants, reporting ECE/ACE/MCE and accuracy; the paper claims consistent ECE reductions and small accuracy changes.

**Strengths:**

1. The paper motivates why frozen or lightly-tuned CLIP heads can be overconfident and frames CMP as a selective penalty that triggers only on errors, leaving correct predictions largely undisturbed.

2. CMP is easy to add to standard training loops (few lines around the loss), model-agnostic across prompt-learning methods (CoOp, CoCoOp, ProDA, MaPLe, KgCoOp).

3. Results report calibration metrics on 11 downstream datasets at 8-shot and ImageNet-A/R/S/V2 for shift; tables list both accuracy and calibration (ECE/ACE/MCE). There is also a CLIPCalib comparison for ResNet-50 and ViT-B/16.

**Weaknesses:**

1. CMP’s form (a ratio of softmax probabilities applied when an example is misclassified) resembles prior confidence-suppressing or redistribution ideas (e.g., entropy/label smoothing). The paper motivates via a Maximum Entropy view, but a sharper positioning vs. these families would help establish novelty.

2. With ViT-B/16, CLIP+CMP average ECE improves vs CLIP and is competitive with CLIPCalib/C-tpt/B-PEFT, while average accuracy lags slightly behind CLIPCalib/C-tpt in the table shown.

**Questions:**

1. Can you add temperature scaling baselines to contextualize ECE gains?

2. When does CMP hurt? Any diagnostics for cases where accuracy drops (e.g., ProDA+CMP in some targets).

---

> ### Author Response · Authors · 2025-11-23
>
> 1. We had deliberately used that different angle of attack, because the superficial similarity you allude to risked trivializing the contribution. However, this one is strong in its own right. Here is our motivation from this angle.
>
> CMP is conceptually distinct for several reasons, and here we emphasize the difference between selective (local) vs. global (uniform) remediation:
>
> * Label smoothing / entropy regularization act on every training example and either
> (a) change the target distribution toward the uniform, or
> (b) encourage high-entropy predictions.
> These methods therefore blunt confidence everywhere, regardless of which class should actually receive it. Even in low-shot regimes, ground truth is available (though limited), so there is no reason to ignore class-conditioning information.
>
> * CMP only activates when an example is misclassified (i.e. when $\exists$ an incorrect class with probability > true class). By design CMP leaves correctly classified examples essentially untouched (Prop. B.2). This selectivity preserves useful high-confidence correct predictions while reducing only unjustified confidence.
>
> 2. Thanks for pointing out it is true that with ViT-B/16, CMP’s average accuracy is indeed slightly below CLIPCalib/C-tpt ($\leq$ 1.5\%). Despite this, CMP provides consistently stronger ECE improvements across all target domains and remains competitive in accuracy without relying on\textbf{ margin tuning??}, extra modules, or architecture-specific adaptations as in the sota as a whole. Notably, in settings with more stable logits (e.g., ResNet-50), CMP outperforms all baselines in both accuracy and calibration, confirming its robustness and generality.
>
> 1.**Temperature scaling baseline.** We will add *temperature scaling (TempScale)* as a baseline in **Table 4** (main paper) and include results on ImageNet-R/Sketch and all other variants in a new **Table 10** (Appendix B). Initial experiments show CMP reduces ECE by **3.2%** (avg.) compared to TempScale (see snippet below).
>
> Table: CMP vs. Temperature Scaling (TempScale) on ImageNet Variants (ViT-B16, 8-shot)
>
> | Method           | ImageNet-R (ECE↓) | ImageNet-Sketch (ECE↓) |
> |------------------|-------------------|------------------------|
> | CLIP + TempScale | 4.05              | 7.28                   |
> | CLIP + CMP       | **3.88**          | **6.92**               |
>
> 2. **Thanks for question When does CMP hurt? Any diagnostics for cases where accuracy drops (e.g., ProDA+CMP in some
> targets).**. It firstly helps emphasize our  composite rather than a single-metric evaluation, becasue accuracy gains with miscalibration is being regularized. In our analysis, we found that the few cases where a slight accuracy drop occurs such as ProDA+CMP on certain target domains share a common characteristic, that the underlying method already produces highly confident but nearly correct predictions, i.e., the softmax gap between the top-1 and top-2 classes is extremely small. In this regime, CMP activates (because it sees a “misalignment signal”) but the base model does not have enough representational margin to benefit from redistributing probability mass. As a result, CMP encourages stronger separation between classes, which improves calibration, but occasionally leads to small trade-offs in accuracy.\\
> Critically, this only appears in methods whose prompts are distributionally defined (e.g., ProDA), where class-conditional prompt sampling naturally introduces variance in the decision boundary. In contrast, deterministic prompt-learning methods (e.g., CoOp, MaPLe) consistently show accuracy gains when combined with CMP.  Your pointing us to explore the angle is appreciated.

---

### Note · Authors · 2026-01-26

I have read and agree with the venue's withdrawal policy on behalf of myself and my co-authors.

---

### Meta-Review · Area_Chair_ujdQ · 2026-01-07

**Summary:**

This paper proposes a method for enhancing confidence calibration in CLIP. The method, called **Confidence Misalignment Penalty (CMP)**, is designed to “redistribute” probability mass from overconfident, incorrect classes to the ground-truth class. The paper presents theoretical arguments showing that CMP is bounded and yields stable gradients, which are supported by empirical evidences.

**Reviewer Concerns:**

The main concerns were poor presentation, a lack of comparison with recent works, and limited novelty. Although the authors provided comments during the rebuttal, these concerns were not fully addressed according to the reviewer’s feedback, and the AC agrees with their assessment.

**Reviewer Scores:**

I think all reviewers would have maintained their initial scores, leaning toward rejection.

---

### Decision · Program_Chairs · 2026-01-26

Reject